

# A machine learning framework for the prediction of chromatin folding in *Drosophila* using epigenetic features

Michal B. Rozenwald[1], Aleksandra A. Galitsyna[2], Grigory V. Sapunov[1,3], Ekaterina E. Khrameeva[2] and Mikhail S. Gelfand[2,4]

[1] Faculty of Computer Science, National Research University Higher School of Economics, Moscow, Russia
[2] Skolkovo Institute of Science and Technology, Moscow, Russia
[3] Intento, Inc., Berkeley, CA, USA
[4] A.A. Kharkevich Institute for Information Transmission Problems, RAS, Moscow, Russia

Corresponding authors
Michal B. Rozenwald,
mbrozenvald@edu.hse.ru,
michal.rozenwald@gmail.com
Mikhail S. Gelfand,
m.gelfand@skoltech.ru,
michal.rozenwald@gmail.com

## ABSTRACT

Technological advances have lead to the creation of large epigenetic datasets, including information about DNA binding proteins and DNA spatial structure. Hi-C experiments have revealed that chromosomes are subdivided into sets of self-interacting domains called Topologically Associating Domains (TADs). TADs are involved in the regulation of gene expression activity, but the mechanisms of their formation are not yet fully understood. Here, we focus on machine learning methods to characterize DNA folding patterns in *Drosophila* based on chromatin marks across three cell lines. We present linear regression models with four types of regularization, gradient boosting, and recurrent neural networks (RNN) as tools to study chromatin folding characteristics associated with TADs given epigenetic chromatin immunoprecipitation data. The bidirectional long short-term memory RNN architecture produced the best prediction scores and identified biologically relevant features. Distribution of protein Chriz (Chromator) and histone modification H3K4me3 were selected as the most informative features for the prediction of TADs characteristics. This approach may be adapted to any similar biological dataset of chromatin features across various cell lines and species. The code for the implemented pipeline, Hi-ChiP-ML, is publicly available: https://github.com/MichalRozenwald/Hi-ChIP-ML

## INTRODUCTION

Machine learning has proved to be an essential tool for studies in the molecular biology of the eukaryotic cell, in particular, the process of gene regulation (*Eraslan et al., 2019*; *Zeng, Wang & Jiang, 2020*). Gene regulation of higher eukaryotes is orchestrated by two primary interconnected mechanisms, the binding of regulatory factors to the promoters and enhancers, and the changes in DNA spatial folding. The resulting binding patterns and chromatin structure represent the epigenetic state of the cells. They can be assayed

by high-throughput techniques, such as chromatin immunoprecipitation (*Ren et al., 2000*; *Johnson et al., 2007*) and Hi-C (*Lieberman-Aiden et al., 2009*). The epigenetic state is tightly connected with inheritance and disease (*Lupiáñez, Spielmann & Mundlos, 2016*; *Yuan et al., 2018*; *Trieu, Martinez-Fundichely & Khurana, 2020*). For instance, disruption of chromosomal topology in humans affects gliomagenesis and limb malformations (*Krijger & De Laat, 2016*). However, the details of underlying processes are yet to be understood.

The study of Hi-C maps of genomic interactions revealed the structural and regulatory units of eukaryotic genome, topologically associating domains, or TADs. TADs represent self-interacting regions of DNA with well-defined boundaries that insulate the TAD from interactions with adjacent regions (*Lieberman-Aiden et al., 2009*; *Dixon et al., 2012*; *Rao et al., 2014*). In mammals, the boundaries of TADs are defined by the binding of insulator protein CTCF (*Rao et al., 2014*). However, *Drosophila* CTCF homolog is not essential for the formation of TAD boundaries (*Wang et al., 2018*). Contribution of CTCF to the boundaries was detected in neuronal cells, but not in embryonic cells of *Drosophila* (*Chathoth & Zabet, 2019*). At the same time, up to eight different insulator proteins have been proposed to contribute to the formation of TADs boundaries (*Ramírez et al., 2018*).

*Ulianov et al. (2016)* demonstrated that active transcription plays a key role in the *Drosophila* chromosome partitioning into TADs. Active chromatin marks are preferably found at TAD borders, while repressive histone modifications are depleted within inter-TADs. Thus, histone modifications instead of insulator binding factors might be the main TAD-forming factors in this organism.

To determine factors responsible for the TAD boundary formation in *Drosophila*, *Ulianov et al. (2016)* utilized machine learning techniques. For that, they formulated a classification task and used a logistic regression model. The model input was a set of ChIP-chip signals for a genomic region, and the output, a binary value indicating whether the region was located at the boundary or within a TAD. Similarly, *Ramírez et al. (2018)* demonstrated the effectiveness of the lasso regression and gradient boosting for the same task.

However, this approach has two substantial limitations. First, the prediction of TAD state as a categorical output depends on the TAD calling procedure. It requires setting a threshold for the TAD boundary definition and it is insensitive to sub-threshold boundaries.

Alternatively, the TAD status of a region may be derived from a Hi-C map either by calculation of local characteristics of TADs such as Insulation Score (*Crane et al., 2015*), D-score (*Stadhouders et al., 2018*), Directionality Index (*Dixon et al., 2012*), or by dynamic programming methods, such as Armatus (*Filippova et al., 2014*). Methods assessing local characteristics of TADs result in assigning a continuous score to genomic bins along the chromosome. Dynamic programming methods are typically not anchored to a local genomic region and consider Hi-C maps of whole chromosomes. The calculation of *transitional gamma* has the advantages of both approaches (*Ulianov et al., 2016*). It runs dynamic programming for whole-chromosome data for multiple parameters and assesses the score for each genomic region.

The second limitation is that regression and gradient boosting in *Ulianov et al. (2016)* and *Ramírez et al. (2018)* account for the features of a given region of the genome, but

ignore the adjacent regions. Such contextual information might be crucial for the TAD status in *Drosophila*.

For a possible solution, one may look at instructive examples of other chromatin architecture problems, such as improvement of Hi-C data resolution (*Gong et al., 2018*; *Schwessinger et al., 2019*; *Li & Dai, 2020*), inference of chromatin structure (*Cristescu et al., 2018*; *Trieu, Martinez-Fundichely & Khurana, 2020*), prediction of genomic regions interactions (*Whalen, Truty & Pollard, 2016*; *Zeng, Wu & Jiang, 2018*; *Li, Wong & Jiang, 2019*; *Fudenberg, Kelley & Pollard, 2019*; *Singh et al., 2019*; *Jing et al., 2019*; *Gan, Li & Jiang, 2019*; *Belokopytova et al., 2020*), and, finally, TAD boundaries prediction in mammalian cells (*Gan et al., 2019*; *Martens et al., 2020*).

The machine learning approaches used in these works include generalized linear models (*Ibn-Salem & Andrade-Navarro, 2019*), random forest (*Bkhetan & Plewczynski, 2018*; *Gan et al., 2019*), other ensemble models (*Whalen, Truty & Pollard, 2016*), and neural networks: multi-layer perceptron (*Gan et al., 2019*), dense neural networks (*Zeng, Wu & Jiang, 2018*; *Farré et al., 2018*; *Li, Wong & Jiang, 2019*), convolutional neural networks (*Schreiber et al., 2017*), generative adversarial networks (*Liu, Lv & Jiang, 2019*), and recurrent neural networks (*Cristescu et al., 2018*; *Singh et al., 2019*; *Gan, Li & Jiang, 2019*).

Among these methods, recurrent neural networks (RNNs) provide a comprehensive architecture for analyzing sequential data (*Graves, Jaitly & Mohamed, 2013*), due to the temporal modeling capabilities. A popular implementation of RNN long short-term memory (LSTM) models (*Hochreiter & Schmidhuber, 1997*) creates informative statistics that provide solutions for complex long-time-lag tasks (*Graves, 2012*). Thus, the application of LTSM RNNs to problems with sequential ordering of a target, such as DNA bins characteristics, is a promising approach. Moreover, this feature is particularly relevant for the TAD boundary prediction in *Drosophila*, where the histone modifications of extended genomic regions govern the formation of boundaries (*Ulianov et al., 2016*).

Here, we analyze the epigenetic factors contributing to the TAD status of the genomic regions of *Drosophila*. As opposed to previous approaches, we incorporate information about the region context on two levels. First, we utilize the context-aware TAD characteristic *transitional gamma*. Second, we use the advanced method of recurrent neural network that preserves the information about features of adjacent regions.

## MATERIALS AND METHODS

### Data

Hi-C datasets for three cultured *Drosophila melanogaster* cell lines were taken from *Ulianov et al. (2016)*. Cell lines Schneider-2 (S2) and Kc167 from late embryos and DmBG3-c2 (BG3) from the central nervous system of third-instar larvae were analysed. The *Drosophila* genome (dm3 assembly) was binned at the 20-kb resolution resulting in 5950 sequential genomic regions of equal size. Each bin was described by the start coordinate on the chromosome and by the signal from a set of ChIP-chip experiments. The ChIP-chip data were obtained from the modENCODE database (*Waterston et al., 2009*) and processed as in *Ulianov et al. (2016)*.

As chromatin architecture is known to be correlated with epigenetic characteristics in *Drosophila* (*Ulianov et al., 2016*; *Hug et al., 2017*; *Ramírez et al., 2018*), we selected two sets of epigenetic marks, i.e., transcription factors (TF), and insulator protein binding sites, and histone modifications (HM), for further analysis. The first set included five features (Chriz, CTCF, Su(Hw), H3K27me3, H3K27ac), which had been reported as relevant for TAD formation in previous studies (*Ulianov et al., 2016*). The second set contained eighteen epigenetic marks in total, extending the first set with thirteen potentially relevant features chosen based on the literature (RNA polymerase II, BEAF-32, GAF, CP190, H3K4me1, H3K4me2, H3K4me3, H3K9me2, H3K9me3, H3K27me1, H3K36me1, H3K36me3, H4K16ac). To normalize the input data, we subtracted the mean from each value and then scaled it to the unit variance using the preprocessing scale function of the Sklearn Python library (*Pedregosa et al., 2011*). We standardized each feature independently; the mean and variance were calculated per each feature (chromatin mark) separately across all input objects (bins), see Fig. S2. For the full list of chromatin factors and their modENCODE IDs, see Table S1.

## Target value

TADs are calculated based on Hi-C interactions matrix. As a result of TAD calling algorithm, TADs are represented as a segmentation of the genome into discrete regions. However, resulting segmentation typically depends on TAD calling parameters. In particular, widely used TAD segmentation software Armatus (*Filippova et al., 2014*) annotates TADs for a user-defined scaling parameter *gamma*. Gamma determines the average size and the number of TADs produced by Armatus on a given Hi-C map.

Following *Ulianov et al. (2016)*, we avoided the problem of selection of a single set of parameters for TADs annotation and calculated the local characteristic of TAD formation of the genome, namely, *transitional gamma*. The calculation of transitional gamma includes the TAD calling for a wide range of reasonable parameters gamma and selection of characteristic gamma for each genomic locus. This procedure is briefly described below.

When parameter gamma is fixed, Armatus annotates each genomic bin as a part of a TAD, inter-TAD, or TAD boundary. The higher the gamma value is used in Armatus, the smaller on average the TADs sizes are. We perform the TAD calling with Armatus for a set of parameters and characterize each bin by transitional gamma at which this bin switches from being a part of a TAD to being a part of an inter-TAD or a TAD boundary. We illustrate the TADs annotation and calculation of transitional gamma in Figs. 1A–1C.

Whole-genome Hi-C maps of *Drosophila* cells were collected from *Ulianov et al. (2016)* and processed using Armatus with a gamma ranging from 0 to 10 with a step of 0.01. We then calculated the transitional gamma for each bin. The resulting distribution of values can be found in Fig. 1D. We note that the value 10 is corresponding to the bins that form TAD regions that we have never observed as being TAD boundary or inter-TAD. These bins might switch from TADs with the further increase of gamma. However, they represent a minor fraction of the genome corresponding to strong inner-TAD bins.

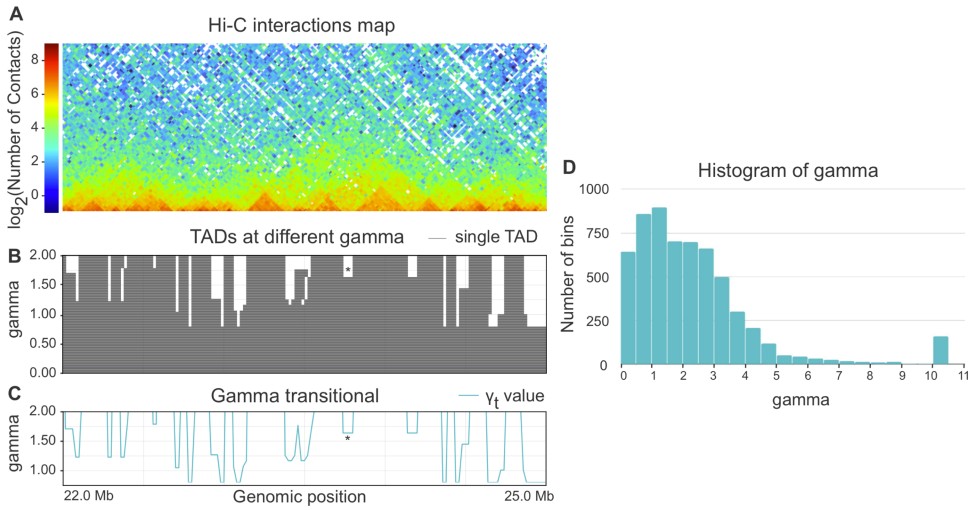

**Figure 1** (A–C) Example of annotation of chromosome 3R region by transitional gamma. For a given Hi-C matrix of Schneider-2 cells (A), TAD segmentations (B) are calculated by Armatus for a set of gamma values (from 0 to 10, a step of 0.01). Each line in B represents a single TAD. Then gamma transitional (C) is calculated for each genomic region as the minimal value of gamma where the region becomes inter-TAD or TAD boundary. The blue line in C represents the transitional gamma value for each genomic bin. The plots (B) and (C) are limited by gamma 2 for better visualization, although they are continued to the value of 10. Asterisk (*) denotes the region with gamma transitional of 1.64, the minimal value of gamma, where the corresponding region transitions from TAD to inter-TAD. (D) The histogram of the target value transitional gamma for Schneider-2 cell line. Note the peak at 10.

## Problem statement

To avoid ambiguity, we formally state our machine learning problem:

- **objects** are genomic bins of 20-kb length that do not intersect,
- **input features** are the measurements of chromatin factors' binding,
- **target value** is the transitional gamma, which characterizes the TAD status of the region and, thus, the DNA folding,
- **objective** is to predict the value of transitional gamma and to identify which of the chromatin features are most significant in predicting the TAD state.

## Selection of loss function

The target, transitional gamma, is a continuous variable ranging from 0 to 10, which yields a regression problem (*Yan & Su, 2009*). The classical optimization function for the regression is *Mean Square Error (MSE)*, instead of precision, recall or accuracy, as for binary variables. However, the distribution of the target in our problem is significantly unbalanced (see Fig. 1D) because the target value of most of the objects is in the interval between 0 and 3. Thus, the contribution of the error on objects with a high true target value may be also high in the total score when using MSE.

We note that the biological nature of genomic bins with high transitional gamma is different from other bins. Transitional gamma equal to 10 means that the bin never transformed from being a part of a TAD to an inter-TAD or TAD boundary. To solve this

contradiction, we have introduced a custom loss function called modified *weighted Mean Square Error (wMSE)*. It might be reformulated as MSE multiplied by the weight (penalty) of the error, depending on the true value of the target variable.

$$wMSE = \frac{1}{N} \sum_{i=1}^{N} (y_{\text{true}_i} - y_{\text{pred}_i})^2 \frac{\alpha - y_{\text{true}_i}}{\alpha},$$

where $N$ is the number of data points, $y_{\text{true}_i}$ is the true value for data point number $i$, $y_{\text{pred}_i}$ is the predicted value for data point number $i$. Here, $\alpha$ is the maximum value of $y_{\text{true}}$ increased by 1 to avoid multiplying the error by 0. The maximum value of the transitional gamma in our dataset is 10, thus in our case, $\alpha$ equals 11. With wMSE as a loss function, the model is penalized less for errors on objects with high values of transitional gamma.

## Machine learning models

To explore the relationships between the 3D chromatin structure and epigenetic data, we built linear regression (LR) models, gradient boosting (GB) regressors, and recurrent neural networks (RNN). The LR models were additionally applied with either L1 or L2 regularization and with both penalties. For benchmarking we used a constant prediction set to the mean value of the training dataset.

Due to the DNA linear connectivity, our input bins are sequentially ordered in the genome. Neighboring DNA regions frequently bear similar epigenetic marks and chromatin properties (*Kharchenko et al., 2011*). Thus, the target variable values are expected to be vastly correlated. To use this biological property, we applied RNN models. In addition, the information content of the double-stranded DNA molecule is equivalent if reading in forward and reverse direction. In order to utilize the DNA linearity together with equivalence of both directions on DNA, we selected the bidirectional long short-term memory (biLSTM) RNN architecture (*Schuster & Paliwal, 1997*). The model takes a set of epigenetic properties for bins as input and outputs the target value of the *middle bin*. The middle bin is an object from the input set with an index $i$, where $i$ equals to the floor division of the input set length by 2. Thus, the transitional gamma of the middle bin is being predicted using the features of the surrounding bins as well. The scheme of this model is presented in Fig. 2.

We exploited the following parameters of the biLSTM RNN in our experiments.

The sequence length of the RNN input objects is a set of consecutive DNA bins with fixed length that was varied from 1 to 10 (*window size*).

The numbers of LSTM units that we tested for were 1, 4, 8, 16, 32, 64, 128, 256, 512.

The weighted Mean Square Error loss function was chosen and models were trained with a stochastic optimizer Adam (*Kingma & Ba, 2014*).

Early stopping was used to automatically identify the optimal number of training epochs. The dataset was randomly split into three groups: train dataset 70%, test dataset 20%, and 10% data for validation.

To explore the importance of each feature from the input space, we trained the RNNs using only one of the epigenetic features as input. Additionally, we built models in which columns from the feature matrix were one by one replaced with zeros, and all other features

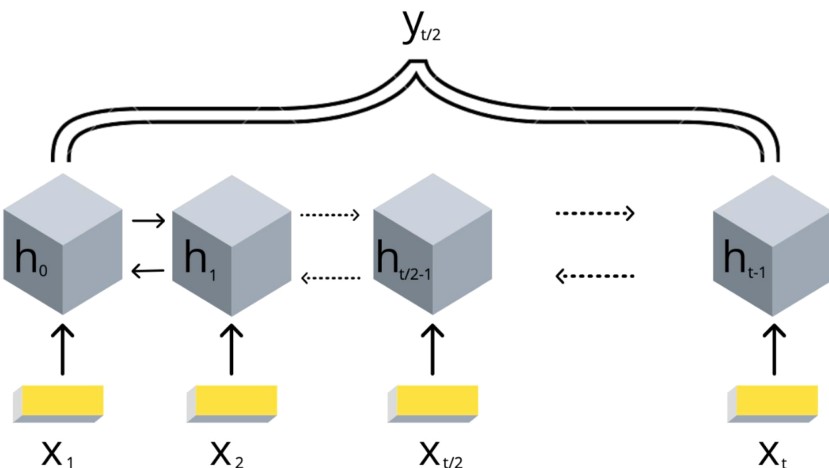

**Figure 2** **Scheme of the implemented bidirectional LSTM recurrent neural networks with one output.** The values of $\{x_1, .., x_t\}$ are the DNA bins with input window size $t$, $\{h_1, .., h_t\}$ are the hidden states of the RNN model, $y_{t/2}$ represents the corresponding target value transitional gamma of the middle bin $x_{t/2}$. Note that each bin $x_i$ is characterized by a vector of chromatin marks ChIP-chip data.

were used for training. Further, we calculated the evaluation metrics and checked if they were significantly different from the results obtained while using the complete set of data.

## RESULTS

### Chromatin marks are reliable predictors of the TAD state

First, we assessed whether the TAD state could be predicted from the set of chromatin marks for a single cell line (Schneider-2 in this section). The classical machine learning quality metrics on cross-validation averaged over ten rounds of training demonstrate strong quality of prediction compared to the constant prediction (see Table 1).

High evaluation scores prove that the selected chromatin marks represent a set of reliable predictors for the TAD state of *Drosophila* genomic region. Thus, the selected set of 18 chromatin marks can be used for chromatin folding patterns prediction in *Drosophila*.

The quality metric adapted for our particular machine learning problem, wMSE, demonstrates the same level of improvement of predictions for different models (see Table 2). Therefore, we conclude that wMSE can be used for downstream assessment of the quality of the predictions of our models.

These results allow us to perform the parameter selection for linear regression (LR) and gradient boosting (GB) and select the optimal values based on the wMSE metric. For LR, we selected alpha of 0.2 for both L1 and L2 regularizations.

Gradient boosting outperforms linear regression with different types of regularization on our task. Thus, the TAD state of the cell is likely to be more complicated than a linear combination of chromatin marks bound in the genomic locus. We used a wide range of variable parameters such as the number of estimators, learning rate, maximum depth of the individual regression estimators. The best results were observed while setting the

**Table 1  Evaluation of classical machine learning scores for all models, based on 5-features and 18-features inputs.**

| Model type | MSE Train | MSE Test | MAE Train | MAE Test | $R^2$ |
|---|---|---|---|---|---|
| Constant prediction | 3.71 | 3.72 | 1.36 | 1.31 | 0 |
| Using 5 features: | | | | | |
| LR + L1 | 2.91 | 2.91 | 1.11 | 1.11 | 0.21 |
| LR + L2 | 2.92 | 2.93 | 1.12 | 1.12 | 0.21 |
| LR + L1 + L2 | 2.86 | 2.87 | 1.11 | 1.11 | 0.23 |
| GB-250 | 2.45 | 2.67 | 1.10 | 1.11 | 0.28 |
| biLSTM RNN | 2.36 | 2.90 | 0.92 | 1.01 | 0.33 |
| Using 18 features: | | | | | |
| LR + L1 | 2.77 | 2.77 | 1.09 | 1.09 | 0.25 |
| LR + L2 | 2.69 | 2.69 | 1.08 | 1.08 | 0.27 |
| LR + L1 + L2 | 2.67 | 2.68 | 1.07 | 1.07 | 0.28 |
| GB-250 | 2.22 | 2.53 | 1.06 | 1.07 | 0.32 |
| **biLSTM RNN** | **2.03** | **2.45** | **0.85** | **0.90** | **0.43** |

**Table 2  Weighted MSE of all models, based on 5-features and 18-features inputs.**

| | 5 features | | 18 features | |
|---|---|---|---|---|
| | Train | Test | Train | Test |
| Constant prediction | 1.61 | 1.62 | 1.61 | 1.62 |
| Linear Regression | 1.20 | 1.20 | 1.13 | 1.14 |
| Linear regression + L1 | 1.17 | 1.17 | 1.12 | 1.12 |
| Linear regression + L2 | 1.18 | 1.19 | 1.11 | 1.12 |
| Linear regression + L1 + L2 | 1.17 | 1.16 | 1.11 | 1.11 |
| Grad boosting 100 estimators | 1.11 | 1.13 | 1.08 | 1.10 |
| Grad boosting 250 estimators | 1.06 | 1.11 | 0.95 | 1.07 |
| **biLSTM 64 units & 6 bins** | **0.83** | **0.88** | **0.79** | **0.84** |

'n_estimators': 100, 'max_depth': 3 and n_estimators': 250, 'max_depth': 4, both with 'learning_rate': 0.01. The scores are presented in Tables 1 and 2.

## The context-aware prediction of TAD state is the most reliable

The alternative model that we studied was biLSTM neural network, which provides explicit accounting for linearly ordered bins in the DNA molecule.

We have investigated the hyperparameters set for biLSTM and assessed the wMSE on various input window sizes and numbers of LSTM units. As we demonstrate in Fig. 3, the optimal sequence length is equal to the input window size 6 and 64 LSTM units. This result has a potential biological interpretation as the typical size of TADs in *Drosophila*, being around 120 kb at 20-kb resolution Hi-C maps which equals to 6 bins.

The incorporation of sequential dependency improved the prediction significantly, as demonstrated by the best quality scores achieved by the biLSTM (Table 2). The selected

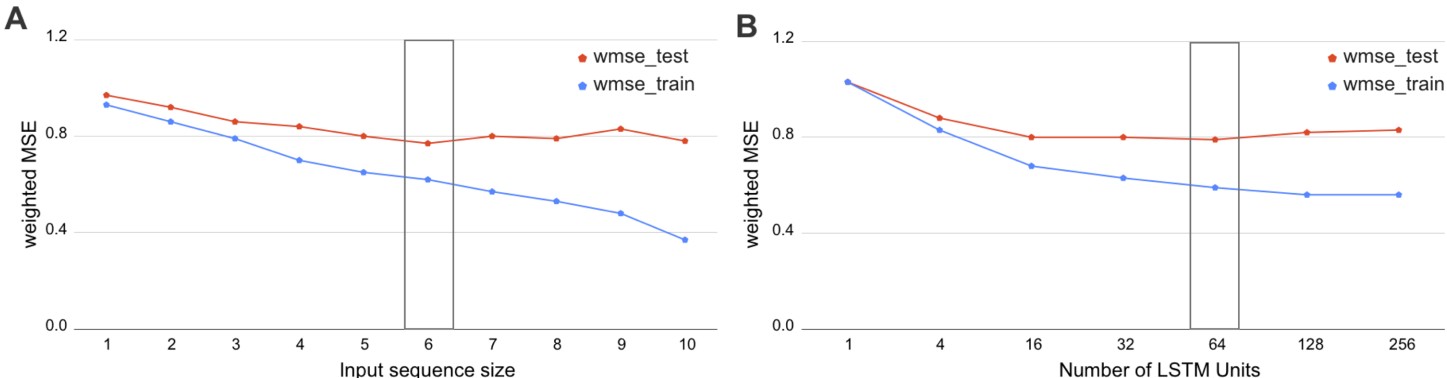

**Figure 3** **Selection of the biLSTM parameters.** Weighted MSE scores for the train and test datasets are presented. (A) Results of RNN with 64 units for different sizes of sequence length. The sequence size corresponds to the input window size of the RNN or number of bins used together as an input sequence for the neural network. (B) Results of RNN with an input sequence of six bins for the different number of LSTM units. The box highlights the best scores. The biLSTM with six input bins and 64 LSTM units was used throughout this study if not specified otherwise.

biLSTM with the best hyperparameters set performed two times better than the constant prediction and outscored all trained LR and GB models, see Tables 1 and 2. We note that the proposed biLSTM model does not take into account the target value of the neighboring regions, both while training and predicting. Our model uses the input values (chromatin marks) solely for the whole window and target values for the central bin in the window for training and assessment of validation results. Thus, we conclude that biLSTM was able to capture and utilize the sequential relationship of the input objects in terms of the physical distance in the DNA.

### Reduced set of chromatin marks is sufficient for a reliable prediction of the TAD state in *Drosophila*

Next, we used an opportunity to analyse feature importance and select the set of factors most relevant for chromatin folding. For an initial analysis, we selected a subset of five chromatin marks that we considered important based on the literature (two histone marks and three potential insulator proteins, 5-features model).

The 5-features model performed slightly worse than the initial 18-features model (see Tables 1 and 2). The difference in quality scores is rather small, supporting the selection of these five features as biologically relevant for TAD state prediction.

We note that the small impact of shrinking of the number of predictors might indicate the high correlation between chromatin features. This is in line with the concept of chromatin states when several histone modifications and other chromatin factors are responsible for a single function of DNA region, such as gene expression (*Filion et al., 2010*; *Kharchenko et al., 2011*).

### Feature importance analysis reveals factors relevant for chromatin folding into TADs in *Drosophila*

We have evaluated the weight coefficients of the linear regression because the large weights strongly influence the model prediction. Chromatin marks prioritization of 5-features LR

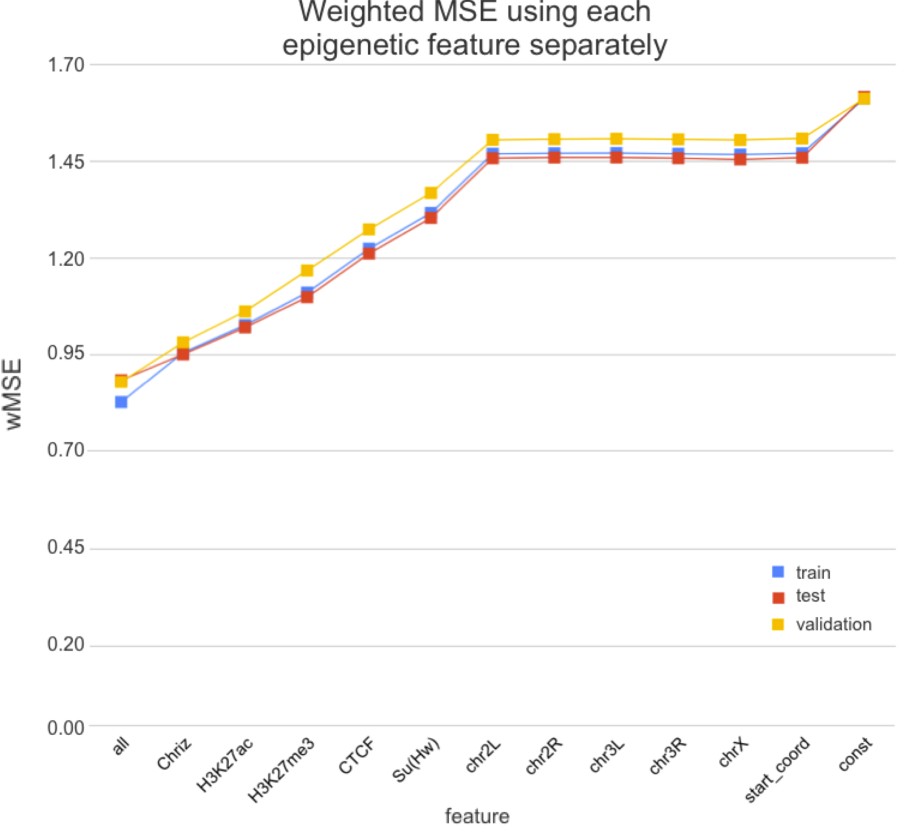

**Figure 4** **Weighted MSE using one feature for each input bin in the biLSTM RNN.** The first mark (‘*all*’) corresponds to scores of NNs using the first dataset of chromatin marks features together, the last mark (‘*const*’) represents wMSE using constant prediction. Note that the lower the wMSE value the better the quality of prediction.

model demonstrated that the most valuable feature was Chriz, while the weights of Su(Hw) and CTCF were the smallest. As expected, Chriz factor was the top in the prioritization of the 18-features LR model. However, the next important features were histone marks H3K4me1 and H3K27me1, supporting the hypothesis of histone modifications as drivers of TAD folding in *Drosophila*.

We used two approaches for the feature selection of RNN: use-one feature and drop-one feature. When each single chromatin mark was used as the only feature of each bin of the RNN input sequence for training, the best scores were obtained for Chriz and H3K4me2 (Figs. 4, 5 and 6), similarly to the LR models results. When we dropped out one of the five features, we got scores that are almost equal to the wMSE using the full dataset together. This does not hold for experiment with excluded Chriz, where wMSE increases. These results align with the outcome of use-one approach and while applying LR models.

Similar results were obtained while using the broader dataset. The results of applying the same approach of omitting each feature one by one using the second dataset of features allowed the evaluation of the biological impact of the features. The corresponding wMSE

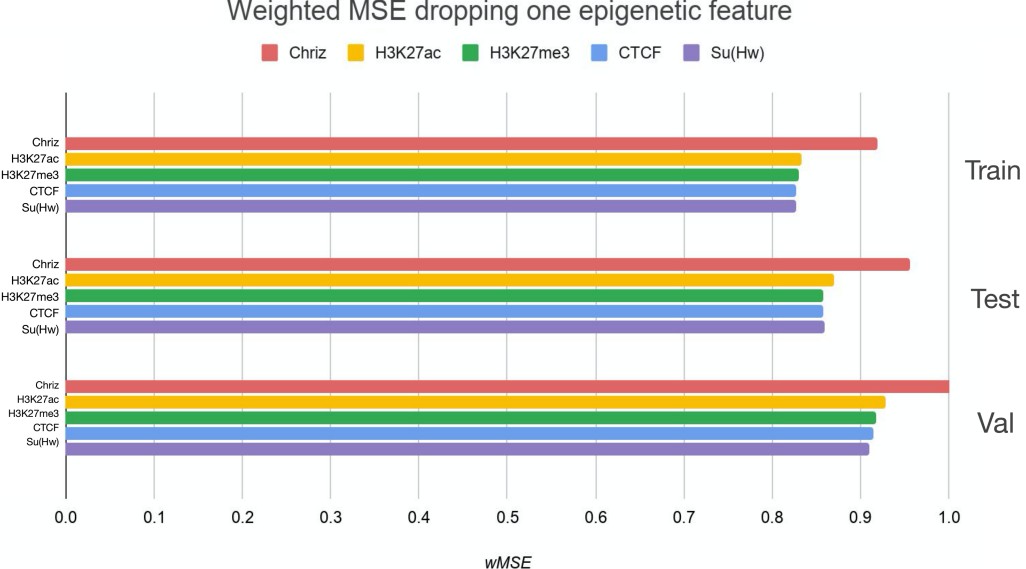

**Figure 5** **Weighted MSE using four out of five chromatin marks features together as the biLSTM RNN input.** Each colour corresponds to the feature that was excluded from the input. Note that the model is affected the most when Chriz factor is dropped from features.

scores are presented in Fig. 6 as well as the result of training the model on all features together.

The results of omitting each feature one by one while using the second dataset of features are almost identical as we expected. It could be explained by the fact that most of the features are strongly correlated.

## TAD state prediction models are transferable between cell lines of *Drosophila*

In order to explore the transferability of the results between various *Drosophila* cell lines, we have applied the full pipeline for Schneider-2 and Kc167 cells from late embryos and DmBG3-c2 (BG3) cells from the central nervous system of third-instar larvae. Across all cell lines, the biLSTM model has gained the best evaluation scores (Table 3). On average, the smallest errors were produced on the test set of the BG3 cell line.

Notably, the selected top features are robust between cell lines. The results of the usage of each feature separately for each of the cell lines can be found in Fig. S1. Chriz was identified as the most influencing feature for Schneider-2 and BG3 while being in the top four features for Kc167. Histone modifications H3K4me2 and H3K4me3 gain very high scores on each dataset. However, CTCF was found in the top of the influencing chromatin marks only on the Kc167, while insulator Su(Hw) constantly scores almost the worst wMSE across all cell lines.

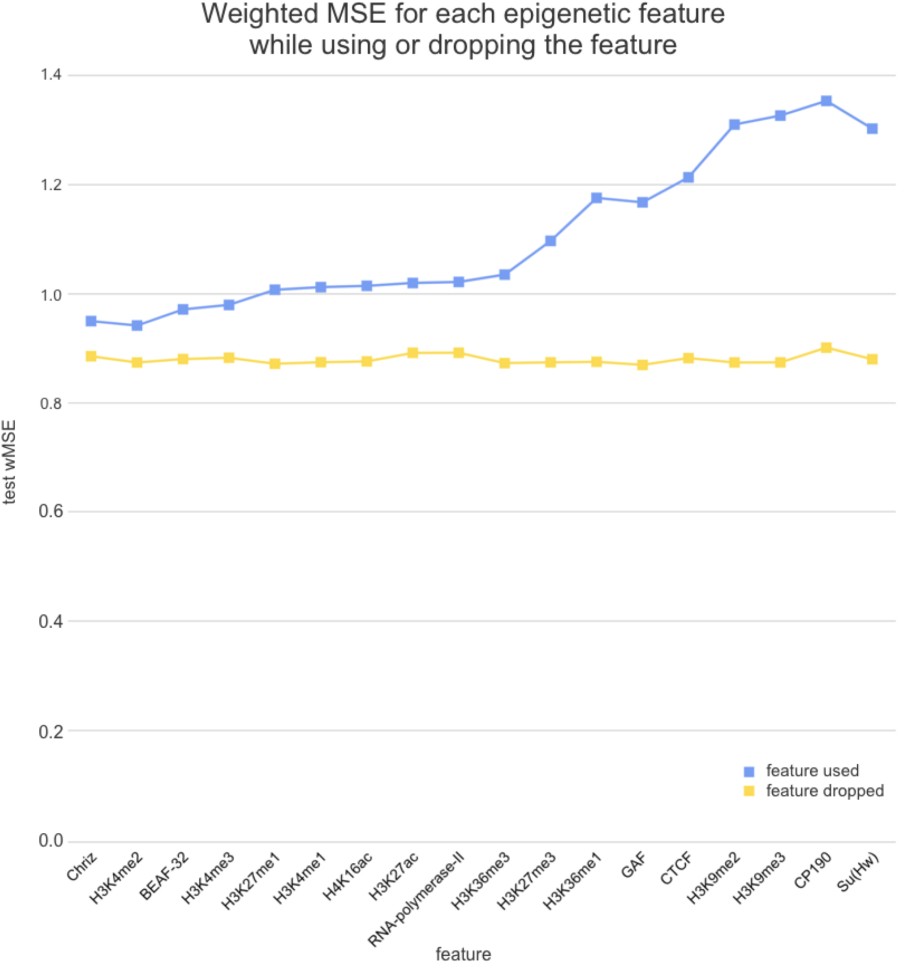

**Figure 6** Weighted MSE on the test dataset while using each chromatin mark either as a single feature (blue line) or excluding it from the biLSTM RNN input (yellow line).

**Table 3** Weighted MSE on cross-validation of all methods for each cell line and while using them together. Lower wMSE orresponds to better quality of prediction.

| Method | Schneider-2 | Kc167 | DmBG3-c2 | All |
|---|---|---|---|---|
| Constant prediction | $1.62 \pm 0.09$ | $1.53 \pm 0.06$ | $1.36 \pm 0.05$ | $1.51 \pm 0.04$ |
| Linear regression | $1.14 \pm 0.08$ | $1.01 \pm 0.06$ | $0.91 \pm 0.08$ | $1.04 \pm 0.04$ |
| Linear regression + L1 | $1.12 \pm 0.07$ | $1.04 \pm 0.06$ | $0.95 \pm 0.07$ | $1.05 \pm 0.04$ |
| Linear regression + L2 | $1.12 \pm 0.07$ | $1.01 \pm 0.06$ | $0.9 \pm 0.08$ | $1.03 \pm 0.04$ |
| Linear regression + L1 + L2 | $1.11 \pm 0.07$ | $1.02 \pm 0.06$ | $0.91 \pm 0.07$ | $1.03 \pm 0.04$ |
| Gradient boosting | $1.07 \pm 0.06$ | $0.98 \pm 0.07$ | $0.86 \pm 0.08$ | $0.96 \pm 0.04$ |
| **biLSTM 64 units & 6 bins** | **$0.86 \pm 0.04$** | **$0.83 \pm 0.04$** | **$0.73 \pm 0.01$** | **$0.78 \pm 0.01$** |

## The all-cell-lines model improves prediction for most cell lines

Finally, we tested the improvement of the prediction models that can be achieved by merging the information about all cell lines. For that, we merged all three cell lines as the input dataset and used the all-cell-lines model for the prediction on each cell line.

The gain of scores was the highest for Schneider-2 and Kc167, while BG3 demonstrated a slight decline in the prediction quality. We also note that biLSTM was less affected by the addition of cross-cell-line data among all models.

In general, the quality of the prediction has mostly improved, suggesting the universality of the biological mechanisms of the TAD formation between three cell lines (two embryonic and one neuronal) of *Drosophila*.

## DISCUSSION

Here, we developed the Hi-ChIP-ML framework for the prediction of chromatin folding patterns for a set of input epigenetic characteristics of the genome. Using this framework, we provide the proof of concept that incorporation of information about the context of genomic regions is important for the TAD status and spatial folding of genomic regions. Our approach allows for diverse biological insights into the process of TAD formation in *Drosophila*, identified using the features importance analysis.

Firstly, we found that chromodomain protein Chriz, or Chromator (*Eggert, Gortchakov & Saumweber, 2004*), might be an important player of the TAD formation mechanism. Recurrent neural networks that used only Chriz as the input produced the highest scores among all RNNs using single epigenetic marks (Figs. 4, 6). Moreover, the removal of Chriz strongly influenced the prediction scores when four out of five selected ChIP features were together (Fig. 5). All linear models assigned the highest regression weight to the Chriz input signal. Further, with the L1 regularization Chriz was the only feature that the model selected for prediction. This chromodomain protein is known to be specific for the inter-bands of *Drosophila melanogaster* chromosomes (*Chepelev et al., 2012*), TAD boundaries and the inter-TAD regions (*Ulianov et al., 2016*), while profiles of proteins that are typically over-represented in inter-bands (including Chriz) correspond to TAD boundaries in embryonic nuclei (*Zhimulev et al., 2014*). The binding sites of insulator proteins Chriz and BEAF-32 are enriched at TAD boundaries (*Hou et al., 2012*; *Hug et al., 2017*; *Ramírez et al., 2018*; *Sexton et al., 2012*). *Wang et al. (2018)* reported the predictor of the boundaries based on the combination of BEAF-32 and Chriz. This might explain BEAF-32 achieving the third rank of the predictability score.

Secondly, the application of the recurrent neural network using each of the selected chromatin marks features separately (Fig. 6) has revealed a strong predictive power of active histone modifications such as H3K4me2. This result aligns with the fact that H3K4me2 defines the transcription factor binding regions in different cells, about 90% of transcription factor binding regions (TFBRs) on average overlap with H3K4me2 regions, and use H3K4me2 together with H3K27ac regions to improve the prediction of TFBRs (*Wang, Li & Hu, 2014*). Histone modifications H3K4me3, H3K27ac, H3K4me1, H3K4me3, H4K16ac, and other active chromatin marks are also enriched in inter-TADs and TAD boundaries (*Ulianov et al., 2016*). In addition, H3K27ac and H3K4me1 distinguish poised and active enhancers (*Barski et al., 2007*; *Creyghton et al., 2010*; *Rada-Iglesias et al., 2011*).

Thirdly, models using Su(Hw) and CTCF perform as expected given that, for the prediction of TAD boundaries, the binding of insulator proteins Su(Hw) and CTCF have

performed worse than other chromatin marks (*Ulianov et al., 2016*). In *Drosophila*, the absence of strong enrichment of CTCF at TAD boundaries and preferential location of Su(Hw) inside TADs implies that CTCF- and Su(Hw)-dependent insulation is not a major determinant of TAD boundaries. Our results also demonstrate that the impact of Su(Hw) and CTCF is low for both proteins.

Thus, our framework not only accurately predicts positions of TADs in the genome but also highlights epigenetic features relevant for the TAD formation. Importantly, the use of adjacent DNA bins created a meaningful biological context and enabled the training of a comprehensive ML model, strongly improving the evaluation scores of the best RNN model.

We note that there are few limitations to our approach. In particular, the resolution of our analysis is 20 kb, while TAD properties and TAD-forming factors can be different at finer resolutions (*Wang et al., 2018*; *Rowley et al., 2017*; *Rowley et al., 2019*). On the other hand, the use of coarse models allowed us to test the approach and select the best parameters while training the models multiple times efficiently. The training of the model for Hi-C with the resolution up to 500 bp presents a promising direction for future work, leading to the clarification of other factors' roles in the formation of smaller TAD boundaries that are beyond the resolution of our models.

We also note that transitional gamma is just one of multiple measures of the TAD state for a genomic region. We motivate the use of transitional gamma by the fact that it is a parameter-independent way of assessing TAD prominence calculated for the entire map. This guarantees the incorporation of the information about the interactions of the whole chromosome at all genomic ranges, which is not the case for other approaches such as the Insulation Score (*Crane et al., 2015*), D-score (*Stadhouders et al., 2018*), and Directionality Index (*Dixon et al., 2012*). On the other hand, the presented pipeline may be easily transferred to predict these scores as target values, which is an important direction for the extension of the work.

Here we selected features that had been reported to be associated with the chromatin structure. We note there might be other factors contributing to the TAD formation that were not included in our analysis. The exploration of a broader set of cell types might be a promising direction for this research, as well as the integration of various biological features, such as raw DNA sequence, to the presented models. We also anticipate promising outcomes of applying our approach to study the chromatin folding in various species except for *Drosophila*.

The code is open-source and can be easily adapted to various related tasks.

## CONCLUSIONS

To sum up, we developed an approach for analysis of a set of chromatin marks as predictors of the TAD state for a genomic locus. We demonstrate a strong empirical performance of linear regression, gradient boosting, and recurrent neural network prediction models for several cell lines and a number of chromatin marks. The selected set of chromatin marks can reliably predict the chromatin folding patterns in *Drosophila*.

Recurrent neural networks incorporate the information about epigenetic surroundings. The highest prediction scores were obtained by the models with the biologically interpretable input size of 120 kb that aligns with the average TAD size for the 20 kb binning in *Drosophila*. Thus, we propose that the explicit accounting for linearly ordered bins is important for chromatin structure prediction.

The top-influencing TAD-forming factors of *Drosophila* are Chriz and histone modification H3K4me2. The chromatin factors that influence the prediction most are stable across the cell lines, which suggests the universality of the biological mechanisms of TAD formation for two embryonic and one neuronal *Drosophila* cell line. On the other hand, the training of models on all cell lines simultaneously generally improves the prediction.

The implemented pipeline called Hi-ChIP-ML is open-source. The methods can be used to explore the 3D chromatin structure of various species and may be adapted to any similar biological problem and dataset. The code is freely available at: https://github.com/MichalRozenwald/Hi-ChIP-ML.

### Funding
This study was supported by the Russian Science Foundation, grant number 19-74-00112, and Skoltech Fellowship in Systems Biology for Aleksandra A. Galitsyna. The funders had no role in study design, data collection and analysis, decision to publish, or preparation of the manuscript.

### Grant Disclosures
The following grant information was disclosed by the authors:
Russian Science Foundation: 19-74-00112.
Skoltech Fellowship in Systems Biology.

### Competing Interests
Mikhail Gelfand is an Academic Editor for PeerJ. Grigory V. Sapunov is employed by Intento, Inc.

### Author Contributions
- Michal B. Rozenwald conceived and designed the experiments, performed the experiments, analyzed the data, performed the computation work, prepared figures and/or tables, authored or reviewed drafts of the paper, and approved the final draft.
- Aleksandra A. Galitsyna conceived and designed the experiments, analyzed the data, prepared figures and/or tables, authored or reviewed drafts of the paper, and approved the final draft.
- Grigory V. Sapunov, Ekaterina E. Khrameeva and Mikhail S. Gelfand conceived and designed the experiments, analyzed the data, authored or reviewed drafts of the paper, and approved the final draft.

## Data Availability

1. The code and the data are available at GitHub: https://github.com/MichalRozenwald/Hi-ChIP-ML

2. The chromatin marks are available at modEncode using the following IDs:

# name Schneider-2 Kc167 DmBG3-c2

1 Chriz 279 277 275
2 CTCF 3749 3749 3671
3 Su(Hw) 5147 3801 3717
4 BEAF-32 922 3745 3663
5 CP190 925 3748 3666
6 GAF 3753 3753 2651
7 H3K4me1 3760 5138 2653
8 H3K4me2 965 4935 2654
9 H3K4me3 3761 5141 967
10 H3K9me2 311 938 310
11 H3K9me3 4183 3013 312
12 H3K27ac 3757 3757 295
13 H3K27me1 3943 3942 3941
14 H3K27me3 298 5136 297
15 H3K36me1 3170 3003 299
16 H3K36me3 303 302 301
17 H4K16ac 320 318 316
18 RNA-polymerase-II 329 328 950

3. The Hi-C data is available at NCBI GEO: GSE69013.

## Supplemental Information

Supplemental information for this article can be found online at http://dx.doi.org/10.7717/peerj-cs.307#supplemental-information.

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
