# Peer review of "A machine learning framework for the prediction of chromatin folding in Drosophila using epigenetic features"

_PeerJ Computer Science, doi:10.7717/peerj-cs.307_

## Round 0.1 · original submission · Minor Revisions

· Academic Editor

Minor Revisions

I believe that you can finish up with the proposed minor revisions n a couple of days.

Reviewer 1 ·

Basic reporting

This is an interesting paper that applies different flavours of machine learning to predict TADs from ChIP-seq data. The Introduction and literature review is well-written. The discussion is well written as well. The Methods section and some figures can be a bit improved. I believe the manuscript needs just a minor revision to address the points below.

Major points:
The methods section needs to be clarified in terms of the experimental data processing. In particular,
1) “To normalize the input data, we mean-centered and scaled each feature to the unit variance” – not clear what exactly was done, please expand.
2) Not clear what is shown on Supplemental Figure 2, even the X and Y axes are not labelled.
3) Please describe in detail the procedure of data normalisation (with equations and/or software tools references when needed).

Minor points:
4) Abstract: “Chriz” -> “Distribution of protein Chriz (Chromator)”.
5) Page 3, line 12: separate “software Armatus” and “Filippova et al.” by some punctuation.

6) Figures 1D, 3 -> increase font sizes so that the smallest font size would be the same as the font size in the main manuscript.
7) Supplementary Figure 2 -> add captions to the X and Y axes and expand figure legend.
8) The first reference in the reference list lacks author names.
9) What is meant by the “sequence size” in Figure 3A? By the way, it would be good to introduce letters “A” and “B” for the two panels in Figure 3.
10) Page 4, line 155: “introduce” -> “Introduced”.
11) Is it possible to say how good the prediction is in terms of standard quantities such as specificity or recall, or anything like this that will tell me whether the wMSE value achieved with the model is really good in comparison with typical performance of other models in the field?

Experimental design

See details in section 1. Basic reporting

Validity of the findings

See details in section 1. Basic reporting

Additional comments

This is an interesting paper that applies different flavours of machine learning to predict TADs from ChIP-seq data. The Introduction and literature review is well-written. The discussion is well written as well. The Methods section and some figures can be a bit improved. I believe the manuscript needs just a minor revision to address the points below.

Major points:
The methods section needs to be clarified in terms of the experimental data processing. In particular,
1) “To normalize the input data, we mean-centered and scaled each feature to the unit variance” – not clear what exactly was done, please expand.
2) Not clear what is shown on Supplemental Figure 2, even the X and Y axes are not labelled.
3) Please describe in detail the procedure of data normalisation (with equations and/or software tools references when needed).

Minor points:
4) Abstract: “Chriz” -> “Distribution of protein Chriz (Chromator)”.
5) Page 3, line 12: separate “software Armatus” and “Filippova et al.” by some punctuation.
6) Figures 1D, 3 -> increase font sizes so that the smallest font size would be the same as the font size in the main manuscript.
7) Supplementary Figure 2 -> add captions to the X and Y axes and expand figure legend.
8) The first reference in the reference list lacks author names.
9) What is meant by the “sequence size” in Figure 3A? By the way, it would be good to introduce letters “A” and “B” for the two panels in Figure 3.
10) Page 4, line 155: “introduce” -> “Introduced”.
11) Is it possible to say how good the prediction is in terms of standard quantities such as specificity or recall, or anything like this that will tell me whether the wMSE value achieved with the model is really good in comparison with typical performance of other models in the field?

Reviewer 2 ·

Basic reporting

No comment.

Experimental design

No comment.

Validity of the findings

No comment.

Additional comments

In the manuscript by Rozenwald et al., a machine learning approach for the prediction of chromatin folding in Drosophila is described. The study is performed by recurrent neural networks.

The manuscript reads well and is very clear. Overall it is a suitable contribution to the journal.

Comments:

1) The Adam optimizer is mentioned on p.5. It will be nice to elaborate a bit and provide a citation.

2) Future work should be added.

---

## Round 0.2 · accepted · Accept

· Academic Editor

Accept

I have carefully read your rebuttal and found it completely satisfactory.